# Modeling Productivity and Estimating Costs of Processor Tower Yarder in Shelterwood Cutting of Pine Stand

**Salvatore F. Papandrea** [1], **Stanimir Stoilov** [2], **Georgi Angelov** [2], **Tanya Panicharova** [2], **Piotr S. Mederski** [3,*] and **Andrea R. Proto** [1]

1. Department of AGRARIA, Mediterranean University of Reggio Calabria, Feo di Vito snc, 89122 Reggio Calabria, Italy
2. Department of Technologies and Mechanization of Forestry, University of Forestry, Kliment Ohridski Blvd. 10, 1756 Sofia, Bulgaria
3. Department of Forest Utilisation, Faculty of Forestry and Wood Technology, Poznan University of Life Sciences (PULS), ul. Wojska Polskiego 71A, 60-625 Poznan, Poland
* Correspondence: piotr.mederski@up.poznan.pl; Tel.: +48-61-848-7761

**Abstract:** Cable-based yarding technology has had a long tradition on steep slopes in Europe, and the new implementation of yarding functions in recent decades favored operational efficiency and lower extraction costs. The main goal of this study was to evaluate the performance of the Syncrofalke 3t truck-mounted Processor Tower Yarder (PTY) on steep terrain, in coniferous forests managed with the shelterwood system. In particular, the aim was to determine PTY productivity and costs, with attention to parameters that could increase PTY effectiveness. The study was carried out in the Sredna Gora Mountains, Central Bulgaria, in pure Scots pine stand, with trees of average DBH = 34 cm and height = 22 m. The study was carried out in six corridors with 120 work cycles of tree extraction up the hill, 28° (53%). The mean productivity of PTY was 15.20 $m^3$ per productive machine hour (PMH) and 12.29 $m^3$ per scheduled machine hour (SMH) and was mainly influenced by the productivity of the yarder unit. Under the given conditions, the performance of PTY significantly increased if more than one tree (at least two trees) were attached and extracted per yarder cycle, since the productivity of the processor was approximately twice that of the yarder. The gross costs of the studied PTY were calculated at 297.48 EUR $PMH^{-1}$ and 16.17 EUR $m^{-3}$. The variable costs (75%) predominate in the net costs distribution, followed by the fixed costs (15%) and the labor costs (10%). The time, productivity and cost results obtained showed the high efficiency and level of integration of PTY operations in order to achieve economic efficiency of logging in montane pine forest managed in a shelterwood system.

**Keywords:** forest operations; cable extraction; wood harvesting; steep terrain

## 1. Introduction

Bulgarian forests are characterized by the small dimensions of most cutting areas, and the predominance of deciduous timber. In the mountain forests of Bulgaria, about 60% are on steep slopes and hence, cable yarders are particularly suitable for timber extraction. In Bulgaria in the 20th century, large areas with coniferous plantations were created at very low altitudes. While coniferous plantations have usually served their primary purpose of helping to control erosion, numerous waves of mortality have been observed in recent decades due to the combined negative effects of drought, aging and lack of opportunities for regular thinnings [1,2]. Generally, 29% of the forests by area are coniferous, but they contribute 45% of growing stock [3]. The coniferous forests offer more options for highly mechanized harvesting technologies, e.g., harvesters and forwarders. The latter, however, have limited mobility and stability—up to slopes of 35%–40%, and the construction of a road network is necessary, requiring significant excavation and embankment works, and thus higher costs, of accessing forest stands.

Cable yarding systems are increasingly being used in all terrains as an alternative to conventional fully mechanized systems with harvesters and forwarders, because of their low impact on soils [4–6] and smaller dependency on slope gradient. Cable yarding can be also used for salvage logging on steep terrains [7] and their safety can be increased with the use of modern anchoring systems [8]. The productivity of cable yarders is strongly influenced by log volume, length of skyline, silvicultural treatment (removal intensity) and lateral yarding distance [9]. In addition, gradient slope, stand density and yarding direction (uphill/downhill) have an influence on the extracting timber volume per unit of time [10–15].

For highly productive harvesting operations on steep terrains based on yarder and mechanized primary tree processing, it is especially important to combine the two operations into one multi-operational machine—Processor Tower Yarders (PTY), representing two independent machines: a yarder and a processor, best if mounted on a single carrier [16]. PTY integrates the drums, a steel spar, power supply, a boom, and a processor head on one carrier [17]. The use of PTY technology is recommended in steep terrain given the improved productivity, which ranges from 90 to 120 $m^3$ per 8 h day [18]. Such technology enables tree processing, sorting, and piling after releasing the load consisting of whole trees [19–21]. Usually, PTY operate in forests affected by windstorms with high removal intensity due to their high productivity, allowing for quick damage coverage. These operating conditions of PTY were studied by Messingerová et al. [22], Bugoš et al. [20], Boyadzhiev and Glushkov [23]. Studies of PTY in thinnings are relatively rare [24]. The processor unit of the tower yarder and its cutting accuracy were typically investigated. Borz et al. [24] examined the performance of the Woody H60 processor of Mounty 4000 without measuring the yarding cycle performance. Marenče et al. [25] measured cutting accuracy with the same processor as part of the Syncrofalke 3t PTY system. The average hourly productivity of one crane processor, as part of PTY, with an average diameter of assortments of 27–28 cm, was 12 to 17 times greater than the productivity of one worker with a chainsaw [23]. The main goal of this study was to evaluate operations of PTY on steep terrain, in coniferous forests managed by the shelterwood system, with regard to economic aspects. Specifically, the objectives were: (*i*) to study the influence of the main operational factors on time consumption of PTY using a statistical modeling approach, and (*ii*) to improve our knowledge on the operational efficiency of PTY in the harvesting operations of coniferous stands managed by the shelterwood system. The study focused on the main operational factors; in this case, on factors during actual extraction of trees and processing of timber, without attention to any additional time and mounting activities preparing PTY for work on the spot. The efficiency improvement concerned the analysis of the time consumption, productivity, and costs, as these performances indicate the level of integration of PTY operations in order to achieve the economic efficiency of timber harvesting in montane coniferous forests.

## 2. Materials and Methods

### 2.1. The Study Site

The study was carried out in the Sredna Gora Mountains (42°39′51.5273″ N–22°20′52.4569″ E) around the city of Koprivshtitsa, Sofia Province, Central Bulgaria (Table 1).

Six parallel-shaped corridors located every about 60 m with an average skyline length of 148 m were opened on terrain slopes at about 27° (51%), 28° (53%), 29° (55%), 27° (51%), 29° (55%), and 29° (55%) (Figure 1). Field observations were carried out on 20 work cycles (turns) at each corridor, with a total of 120 work cycles (turns). During the operations, the extraction direction was uphill, the single-span layout of the cable yarder unit was implemented each time and trees were felled manually with a chainsaw.

**Table 1.** Characteristics of the test site.

| Parameter | Characteristics |
|---|---|
| Place Name | Sub-compartment 9009-l |
| Elevation | 1100 m asl |
| Function | Natura 2000: BG 0001389, BG 0002054 |
| Species composition | Scots pine (*Pinus sylvestris* L.) |
| Stand age | 70 years |
| Stand type | Forest plantation |
| Total area | 6.0 ha |
| Relative stocking | 0.7 |
| Sylvicultural system | Combined regular and shelterwood cut, removal intensity 25% |
| Average tree height | 22 m |
| Average DBH of tree | 34 cm |
| Average slope gradient | 28° (53%) |
| Growing stock | 1794 m$^3$ (299 m$^3$ ha$^{-1}$) |
| Allowable cut | 470 m$^3$ (78 m$^3$ ha$^{-1}$) |
| Extraction direction | Uphill |
| Length of line in corridors | 1: 80 m; 2: 160 m<br>3: 160 m; 4: 150 m<br>5: 150 m; 6: 185 m |
| Average lateral yarding distance | 14.78 m |

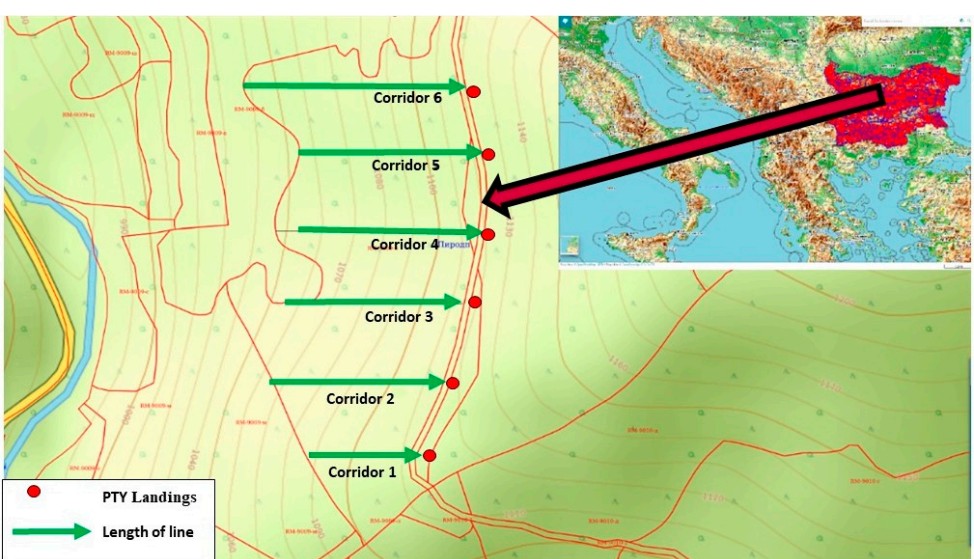

**Figure 1.** Site and yarding corridors.

## 2.2. Cable Yarder Unit

Within the study, a Syncrofalke 3t truck-mounted processor tower yarder (Mayr–Melnhof Forsttechnik GmbH, Frohnleiten, Austria, Table 2) was tested. The work team consisted of two men, of which one was the winch and processor operator at the landing, and the second was the choker setter in the stand. Each operator had at least 5 years of experience with cable yarding, and they were both 35–40 years old.

**Table 2.** Technical data of the studied Syncrofalke 3t processor tower yarder.

| Parameter | Value |
|---|---|
| **Tower yarder** | |
| Skyline capacity | 750 m, ø20 mm |
| Mainline | 1800 m, ø11 mm |
| Haulback line | 1800 m, ø8 mm |
| Guylines | 4 × 70 m, ø18 mm |
| Foldable telescopic tower, height | 11.5 m |
| Power station | Truck engine and hydrostatic transmission |
| Engine power of the truck engine | 324 kW |
| Carriage | MM-Sherpa U3 active slack-pulling carriage |
| Choker system | Bardon choker |
| Hydraulic crane | Palfinger Epsilon S280L94—reach of 9.4 m and a lifting moment of 229 kNm at a working pressure of 250 bar. |
| **Processor** | |
| Processor head | Woody H60 |
| Delimbing diameter | 8–65 cm |
| Max. grapple opening | 120 cm |
| Feed force | 35 kN |
| Weight | 1.450 kg |
| Operating pressure | 300–350 bar |
| Chain speed | 40 m/s |
| Length of the saw guide bar | 820 mm |
| Max. cutting diameter | 680 mm |
| Chain pitch | 0.404″ |
| Number of drive links | 98 |
| Carrier | 6 × 4 Iveco, model 410 Trakker |

The tested PTY is designed for all-terrain harvesting, mounted on a truck with pressure air brakes. The Syncrofalke 3t (Figure 2) has a powerful yarder unit, principally used for selective cutting and for regenerative harvesting operations using a carriage Sherpa U3 (Mayr–Melnhof Forsttechnik GmbH, Frohnleiten, Austria) for payloads up to 3t. The processing of felled trees was carried out on the site with the Woody H 60 processing head (Konrad Forsttechnik GmbH, Preitenegg, Austria), mounted on the Palfinger Epsilon (Epsilon Kran GmbH, Elsbethen-Glasenbach, Austria) model S280L94 hydraulic crane. The trees were yarded laterally to the carriage using the power of the yarder's mainline winch and active skyline clamps [26,27].

### 2.3. Productivity and Costs

A time and motion study were carried out to estimate the duration of work elements and productivity of the cable yarders in the given conditions. A yarding work cycle was assumed to be composed of repetitive elements [13,26–30]. In this study, six (1–6) work elements were separated and taken into account in order to estimate the yarder work cycle time and four for processor unit (7–10) [31]; they were similar to those described by Proto and Zimbalatti [26]:

(1) Carriage outhaul (CO): begins when the operator is ready to move the empty carriage from the landing out to the stump and ends when the choker setter touches the chokers;

(2) Lateral outhaul and hook (LOH): begins at the end of carriage outhaul and ends when the choker setter has completed hooking the chokers and signals to begin yarding;

(3) Lateral inhaul (LI): begins at the end of the hook up and ends when the turn is pulled up to the carriage and the carriage begins to move up the corridor;

(4) Carriage inhaul (CI): begins at the end of lateral inhaul and ends when the load has reached the deck, where it can be directly unhooked at the landing;

(5) Unhook (UN): begins at the end of carriage inhaul and ends when the chokers have returned to the carriage;

(6) Delay time of tower yarder unit (DELy): includes the rest, personal delays, organizational delays, service, and repair.

(7) Direction and gripping (DG): directing and gripping the tree with the processor head.

(8) Delimbing and bucking of the tree (DB): begins after aiming the processor and taking the tree off the landing.

(9) Sorting, piling (SP): after delimbing the tree, the sorting and piling of the woody assortments takes place, as well as the cleaning of any debris.

(10) Delay time of processor unit (DELp): includes rest, personal delays, organizational delays, service, and repair.

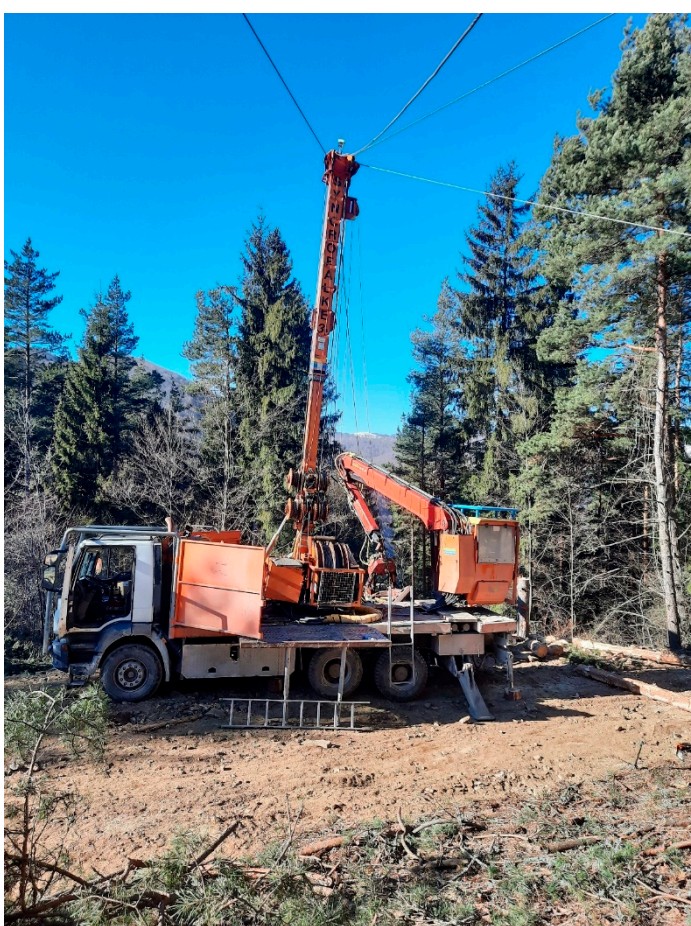

**Figure 2.** Syncrofalke 3t Processor Tower Yarder.

The time-motion study of both units of PTY was designed to evaluate the duration of work elements and the productivity of the yarder and the processor, and to identify those variables that are most likely to affect it. Each yarding cycle and each processor cycle were individually measured using a stopwatch and the productive time was separated from the

delay time. The yarding distances and terrain slopes were measured with a professional laser range finder with a clinometer. The cycle load volume turn was determined by measuring the diameter at breast height and height of the tree and calculated by using biometric models of Nedyalkov et al. [32].

The machine costs were calculated using the COST model [33]. In order to calculate the production cost per 1 m$^3$ of timber, the cost analysis employed the following parameters: the number of operators, the hourly cost of an operator, the hourly cost of machines, the volume of extracted timber and the productive machine hours (PMHs), excluding all delay times. The machine costs per hour were reported as both PMHs and scheduled machine hours (SMHs). The purchase prices and operator wages required by the cost calculations were obtained from catalogues and accounting records [34]. The diesel fuel consumption was measured by evaluating the volume of fuel used to fill the fuel tank to the brim and recording the amount of fuel used during that day. A salvage value of 10% of the purchase price was assumed, and the Value Added Tax (VAT) was excluded. Cost calculations were based on the assumption that companies worked for 200 working days in the year and a depreciation period of 10 years. For extraction work, this amounts to 130–150 working days per year (20–21 working days per month), at an average of 6–7 scheduled working hours per day (assuming one to two hours spent on lunch, rest and other breaks). This yielded annual working times of 910–1050 SMH with a 70% use coefficient [26,34].

### 2.4. Data Analysis

A regression analysis was performed on the experimental data to obtain prediction models for estimating the work cycle time and productivity. The independent variables used in the modeling approach of the yarder unit and PTY included yarding distance *L*, lateral yarding distance *l*, load volume per cycle *V*, terrain slope angle *i* and the load's number of trees *n*, whereas the independent variable used in the modeling of the processor unit performance was load volume per cycle *V*. The descriptive statistics of the variables were computed, and a stepwise backward regression procedure was used to model the variability of yarding cycle time and productivity as a function of independent variables.

The confidence level used for regression analysis was 95% ($\alpha = 0.05$) and the assumed probability $p < 0.05$. Independent variables are significant at $p < 0.05$, i.e., strong presumption against neutral hypothesis. The experimental data were processed by Statistica 8 (StatSoft Inc., Tulsa, OK, USA) software.

## 3. Results and Discussion

There were experimental data from 120 yarding and processor cycles for each of the selected variables used in the cycle time (Table 3). Based on this, production equations were developed (Table 4). During each yarding cycle, one tree was extracted, which was further processed into logs on the landing site.

### 3.1. Duration of Work Cycle Elements
#### 3.1.1. Tower Yarder Unit

The predominant part of the cycle time (Figure 3) was dedicated to the lateral outhaul and hook the tree (25% and 17%, respectively, excluding and including delays), followed by lateral inhaul (22% and 15%, respectively, excluding and including delays) and carriage inhaul by approximately the same proportion (22% and 15%, respectively, excluding and including delays). The other working elements have the following partitions in the cycle: carriage outhaul (16% and 12%, respectively, excluding and including delays), unhook (15% and 10%, respectively, excluding and including delays). Operational and mechanical delays accounted, respectively, for 26% and 5% of the total cycle time of the studied cable yarder unit.

**Table 3.** Mean experimental data.

| Variables | Cycle Time, s | | | Distance, m | | |
|---|---|---|---|---|---|---|
| | Mean Value $\pm$ St. dev. | Min | Max | Mean Value $\pm$ St. dev. | Min | Max |
| **Yarding** | | | | | | |
| Carriage Outhaul (CO) | 31.58 $\pm$ 11.09 | 14 | 62 | 67.44 $\pm$ 15.36 | 40 | 85 |
| Lateral outhaul and hook (LOH) | 47.21 $\pm$ 11.77 | 25 | 79 | 14.78 $\pm$ 3.77 | 7 | 22 |
| Lateral Inhaul (LI) | 41.64 $\pm$ 11.18 | 25 | 69 | 14.78 $\pm$ 3.77 | 7 | 22 |
| Carriage Inhaul (CI) | 41.13 $\pm$ 10.34 | 18 | 62 | 67.44 $\pm$ 15.36 | 40 | 85 |
| Unhook (U) | 28.22 $\pm$ 3.69 | 19 | 39 | | | |
| Delays (DELy) | 82.70 $\pm$ 120.18 | 0 | 660 | | | |
| Total cycle time | 272.48 $\pm$ 121.70 | 151 | 832 | | | |
| Delay-free cycle time | 189.78 $\pm$ 26.52 | 151 | 262 | | | |
| Load volume per cycle (turn), m$^3$ | 1.23 $\pm$ 0.33 | 0.55 | 2.63 | | | |
| Productivity, m$^3$ per PMH | 23.84 $\pm$ 7.52 | 11.75 | 58.09 | | | |
| Productivity, m$^3$ per SMH | 18.41 $\pm$ 6.20 | 5.19 | 58.09 | | | |
| Number of cycles per SMH | 14.97 $\pm$ 4.57 | 4.33 | 23.84 | | | |
| **Tree Processing** | | | | | | |
| Directing and gripping the tree with the processor head (DG) | 16.15 $\pm$ 2.88 | 11 | 26 | | | |
| Delimbing and bucking of the tree (DB) | 62.10 $\pm$ 8.47 | 46 | 84 | | | |
| Sorting, piling the assortments and clearing the debris (SP) | 24.27 $\pm$ 3.54 | 17 | 36 | | | |
| Delays (DELp) | 8.32 $\pm$ 74.95 | 0 | 305 | | | |
| Total cycle time | 110.83 $\pm$ 74.95 | 74 | 103 | | | |
| Delay-free cycle time | 102.52 $\pm$ 5.81 | 74 | 389 | | | |
| Productivity, m$^3$ per PMH | 42.71 $\pm$ 7.74 | 26.76 | 75.13 | | | |
| Productivity, m$^3$ per SMH | 40.89 $\pm$ 9.26 | 8.42 | 75.13 | | | |
| Number of cycles per SMH | 34.18 $\pm$ 4.14 | 24 | 50 | | | |
| **Processor Tower Yarder** | | | | | | |
| Total cycle time | 383.83 $\pm$ 123.98 | 244 | 929 | | | |
| Delay-free cycle time | 292.29 $\pm$ 30.51 | 239 | 399 | | | |
| Productivity, m$^3$ per PMH | 15.20 $\pm$ 3.97 | 8.18 | 32.00 | | | |
| Productivity, m$^3$ per SMH | 12.29 $\pm$ 3.98 | 4.67 | 26.52 | | | |

St. dev.—standard deviation, PMH—productive machine hour, SMH—scheduled machine hour.

Operations related to the lateral yarding (the lateral pull to the main line, the chokers hooking, and the extraction of the load to carriage) occupied 27% of the work cycle time including delays, and 40% of the delay-free work cycle. In the given conditions, the tower yarder operated at relatively short yarding distance (mean 67.44 m out of nominal length), moderate lateral yarding distance (mean 14.78 m) and slope (mean 28.2°) and relatively low level of the carriage payload capacity usage. Lateral yarding had a great influence on the duration of the work cycle.

A regression analysis was performed on the time-study data using characteristics of independent variables (Table 3) in order to develop a prediction equation for estimating the yarding cycle time by excluding and including delays.

**Table 4.** Summary of the work cycle time models ($T_{net}$).

| Equations | | $F$ | $R^2$ | $R^2_{adj}$ | SE | *p*-Value |
|---|---|---|---|---|---|---|
| $T_{net\_Y} = 4.34 \cdot i + 0.91 \cdot L + 3.88 \cdot l$ | **(1)** | **36.71** | **0.56** | **0.55** | **18.03** | **<0.05** |
| $T_{net\_Y\_1} = 0.99 \cdot L + 6.017 \cdot l$ | | 24.05 | 0.82 | 0.78 | 12.09 | <0.05 |
| $T_{net\_Y\_2} = 66.10 + 0.86\,L + 3.64\,l$ | | 15.76 | 0.58 | 0.53 | 15.76 | <0.05 |
| $T_{net\_Y\_3} = 53.43 + 0.99\,L + 3.97\,l$ | | 24.34 | 0.74 | 0.71 | 13.37 | <0.05 |
| $T_{net\_Y\_4} = 158.09 + 0.90\,L + 3.80\,l - 65.85\,V$ | | 18.99 | 0.78 | 0.74 | 14.45 | <0.05 |
| $T_{net\_Y\_5} = 150.13 + 0.76\,L + 3.36\,l - 41.51\,V$ | | 16.51 | 0.76 | 0.71 | 12.31 | <0.05 |
| $T_{net\_Y\_6} = 92.86 + 1.07\,L + 2.67\,l$ | | 23.30 | 0.72 | 0.69 | 11.93 | <0.05 |
| $T_{net,P} = 16.58 + 21.26\,V$ | **(2)** | **173.57** | **0.60** | **0.59** | **8.14** | **<0.05** |
| $T_{net\_P\_1} = 60.52 + 29.40\,V$ | | 62.59 | 0.78 | 0.76 | 5.58 | <0.05 |
| $T_{net\_P\_2} = 64.04 + 25.45\,V$ | | 130.13 | 0.88 | 0.87 | 5.44 | <0.05 |
| $T_{net\_P\_3} = 65.82 + 27.99\,V$ | | 62.48 | 0.78 | 0.76 | 4.25 | <0.05 |
| $T_{net\_P\_4} = 48.38 + 44.55\,V$ | | 37.11 | 0.67 | 0.66 | 4.22 | <0.05 |
| $T_{net\_P\_5} = 54.17 + 44.26\,V$ | | 63.28 | 0.78 | 0.77 | 4.68 | <0.05 |
| $T_{net\_P\_6} = 66.66 + 37.22\,V$ | | 23.82 | 0.56 | 0.53 | 5.64 | <0.05 |
| $T_P = 83.00 + 22.65\,V.$ | **(3)** | **6.44** | **0.05** | **0.04** | **32.32** | **<0.05** |
| $T_{net,PTY} = 8.28 \cdot i + 0.90 \cdot L + 3.94 \cdot l + 20.81\,V$ | **(4)** | **155.56** | **0.60** | **0.60** | **6.11** | **<0.05** |
| $T_{net\_PTY\_1} = 1.082 \cdot L + 6.58 \cdot l + 49.68\,V$ | | 17.46 | 0.77 | 0.72 | 14.67 | <0.05 |
| $T_{net\_PTY\_2} = 132.22 + 0.89\,L + 3.48\,l + 24.01\,V$ | | 13.83 | 0.72 | 0.67 | 15.76 | <0.05 |
| $T_{net\_PTY\_3} = 97.29 + 0.99\,L + 4.19\,l + 42.20\,V$ | | 20.75 | 0.80 | 0.76 | 13.18 | <0.05 |
| $T_{net\_PTY\_4} = 174.88 + 0.97\,L + 3.88\,l$ | | 20.48 | 0.71 | 0.67 | 15.39 | <0.05 |
| $T_{net\_PTY\_5} = 210.42 + 0.69\,L + 3.50\,l$ | | 14.52 | 0.63 | 0.59 | 12.94 | <0.05 |
| $T_{net\_PTY\_6} = 187.43 + 1.20\,L + 3.46\,l$ | | 18.28 | 0.67 | 0.63 | 15.75 | <0.05 |

Y—tower yarder unit, P—processor unit, PTY—processor tower yarder.

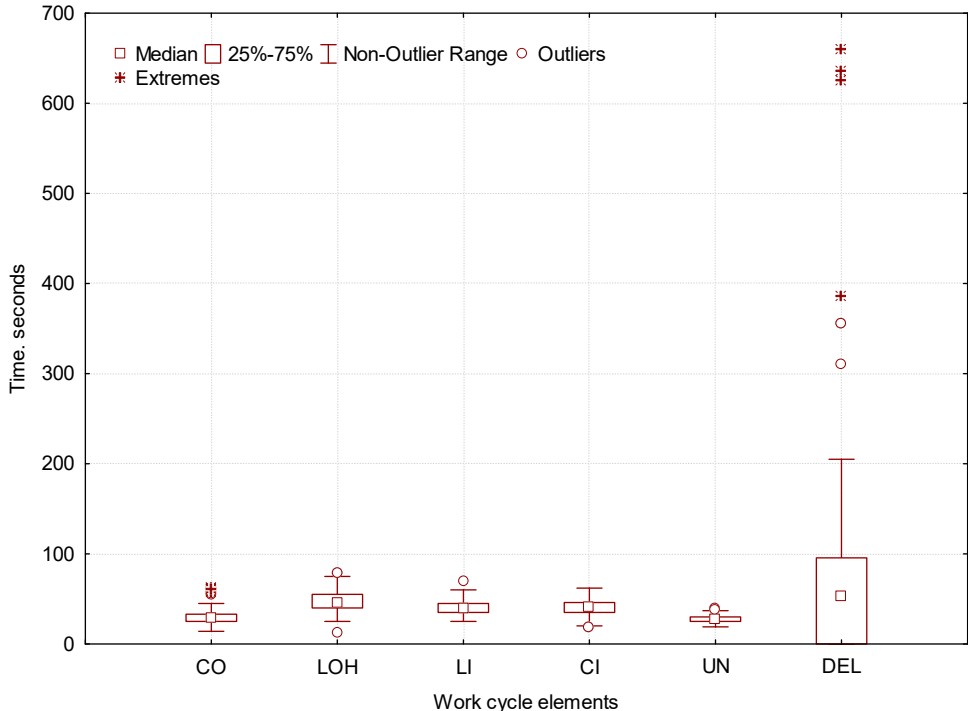

**Figure 3.** Elemental time consumption of tower yarder unit.

The delay-free yarder cycle time $T_{net,Y}$ regression model (1) obtained with significant variables (Table 4), in particular, for each corridor has been determined using a statistical equation to support the results obtained. In Equation (1), minimum values of $T_{net,Y}$ may be obtained in case of lower rates of yarding distance L, lateral yarding distance l and terrain slope angle i.

In corridors 1, 2, 3 and 6, the significant factors that determined the duration of the delay-free work cycle were the yarding distance and the lateral yarding distance, while in corridors 4 and 5, the load, i.e., the volume of the tree, also had an influence. This was probably due to the reduced variation in load volume in corridors 1, 2, 3 and 6.

### 3.1.2. Processor Unit

Analyzing the work cycle of the processor unit, it can be seen that the longest was the time for delimbing and cross-cutting (61% and 56%, respectively, excluding and including delays), followed by the time for sorting, piling and clearing (23% and 22%, respectively, excluding and including delays). The shortest time was for directing and gripping (16% and 15%, respectively, excluding and including delays). The delays comprised 15% of the total cycle time.

Generally, the mean duration of the total cycle time of the processor was 110.83 s and took place during the time of the yarder's next work cycle (mean 272.48 s). Under the given forest conditions, according to Equation (2) for delay-free processor cycle time $T_{net,P}$ and Equation (3) for the processor cycle time, including delays $T_P$, the minimum duration was achieved when load volume V was minimized (Table 4 and Figure 4).

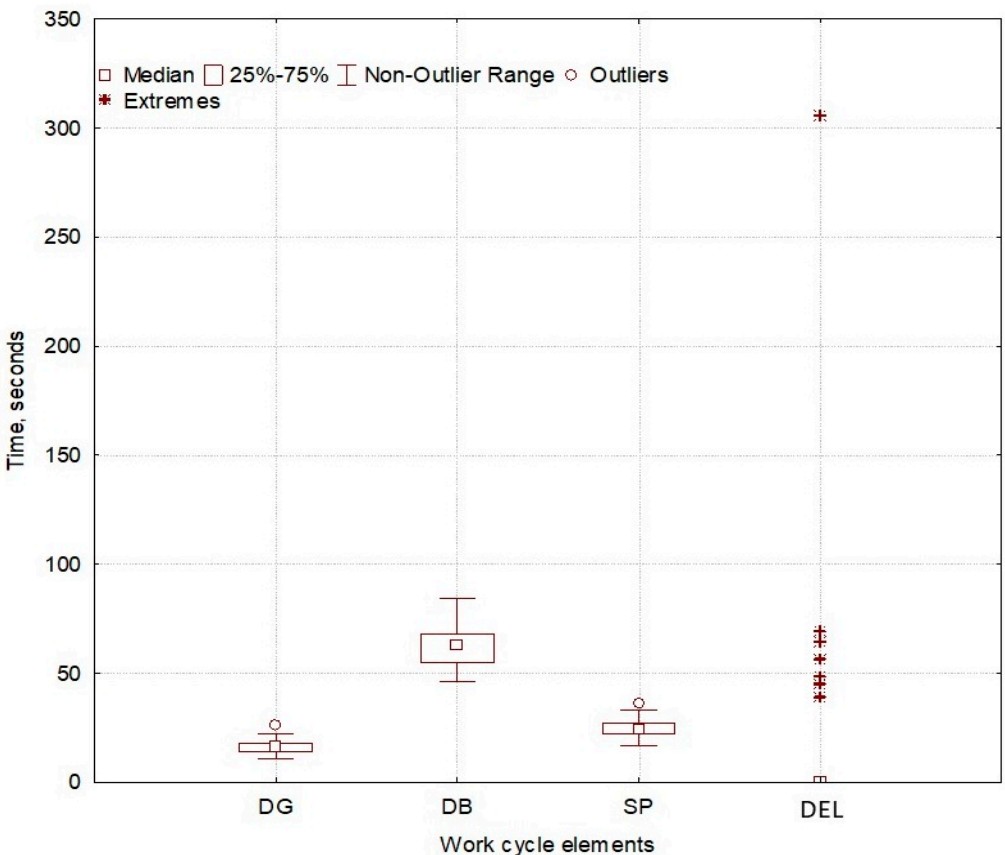

**Figure 4.** Elemental time consumption of processor unit.

Comparing the work cycle times of the yarder and processor, it can be concluded that under the given conditions, the processor usually had enough time to process two trees instead of one, which was what the carriage load consisted of.

### 3.1.3. Processor Tower Yarder

The analysis of the delay-free work cycle of PTY as a multi-operational machine showed the following distribution of work elements in yarding and processing a tree: the longest was delimbing and bucking (21%), followed by outhaul and hooking (16%), load outhaul and carriage inhaul (14% each), carriage outhaul (11%), unhooking (10%), sorting,

piling and clearing (8%) and directing and gripping the tree (6%) (Figure 5). This was similar to the work cycle percentage distribution of the Mounty 4000 PTY, even though research was carried out in spruce stands affected by outbreak of bark beetle and fungi [20]. Similar work cycle proportions when Mounty 4000 was used were also found in oak stand [22]; however, tree trunks were processed after delimbing with chainsaw. Cross-cutting though, was recognized as difficult due to broadleaved tree species, and hard in this case [22].

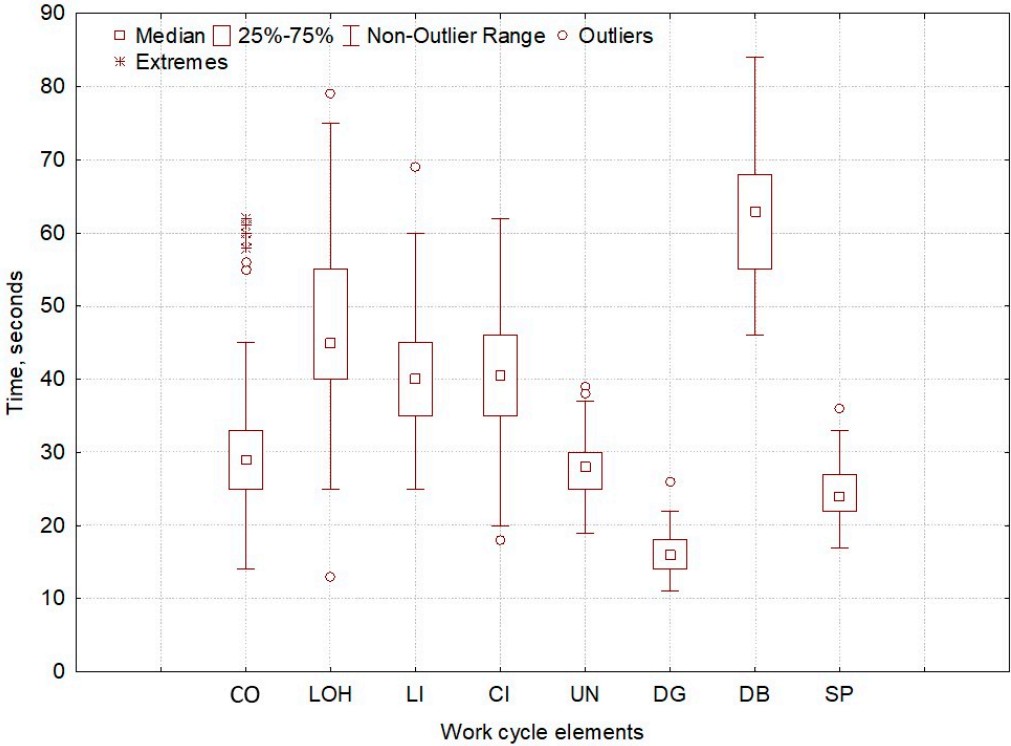

**Figure 5.** Elemental time consumption of processor tower yarder.

The mean duration of the delay-free work cycle of PTY in given operational conditions was 292.13 s. According to Equation (4) describing the delay-free work cycle time $T_{net,PTY}$ (Table 4), a minimum duration can be expected at minimum values of all independent variables: yarding distance L, lateral yarding distance l, terrain slope angle i and load volume V.

*3.2. Productivity*

3.2.1. Tower Yarder Unit

To increase delay-free yarding productivity, defined by Equation (5), yarding distance L, lateral yarding distance l, and terrain slope i should be at low rates, whereas the load volume V per cycle will be at its maximum (Table 5).

From Equation (6), it can be seen that when increasing the volume of a load V to the allowed maximum, it could be expected that the yarding productivity per scheduled machine hour PSMH,Y will increase its maximum (Table 5).

Generally, the mean yarding productivity per hour at a mean slope yarding distance of 67.44 m and mean lateral yarding distance of 14.78 m, excluding and including delays, is estimated at 23.84 m$^3$ PMH$^{-1}$ and 18.41 m$^3$ SMH$^{-1}$ at the given operating conditions. The mean yarding productivity per hour compared to the rate of a tower cable yarder in a salvage operation in the same region for an extraction distance of 101 m and a lateral yarding distance of 18 m, resulting in a productivity rate of 20.1 m$^3$ PMH$^{-1}$ and 12.8 m$^3$ SMH$^{-1}$, including delays, was very close [35]. This can be seen as good result in presented research with shalterwood cutting, as in salvage logging, all affected trees are harvested and extracted, and in the compared case, this occurred for 1/3 of them. Comparison to

the results obtained in thinnings in spruce stand with Mounty 4100 by Borz et al. [24,36] show 5.88 m$^3$ PMH$^{-1}$ at average lateral yarding distance of 22 m and yarding distance of 190.13 m, i.e., more than three times lower rates. This is natural, as in thinnings, smaller trees are usually harvested. In this case they were of 0.308 m$^3$ in average [35]. Additionally, the yarding distance was nearly three times longer when thinning was applied: 190.13 m compared with 67.44 m in the presented study. In order to improve the yarder productivity, the load volume (i.e., the number of trees) of the carriage is advisable to be increased, despite the increase in the lateral yarding time.

**Table 5.** Summary of the productivity (*P*) models.

| Equations | | *F* | *R*$^2$ | *R*$^2_{adj}$ | SE | *p*-Value |
|---|---|---|---|---|---|---|
| **$P_{PMH,Y}= 29.89 − 0.65·i − 0.10·L − 0.44·l + 20.92·V$, m$^3$·h$^{-1}$** | **(5)** | **232.87** | **0.89** | **0.89** | **2.65** | **<0.05** |
| $P_{PMH\_Y\_1} = 16.45 − 0.12·L − 0.44·l + 18.54·V$ | | 406.41 | 0.99 | 0.98 | 1.02 | <0.05 |
| $P_{PMH\_Y\_2} = 15.55 − 0.13 L − 0.49 l + 20.67 V$ | | 93.03 | 0.95 | 0.94 | 3.01 | <0.05 |
| $P_{PMH\_Y\_3} = 23.96 − 0.13 L − 055 l + 14.98 V$ | | 48.49 | 0.90 | 0.88 | 1.91 | <0.05 |
| $P_{PMH\_Y\_4} = − 0.090 L − 0.39 l + 26.23 V$ | | 40.04 | 0.88 | 0.86 | 1.80 | <0.05 |
| $P_{PMH\_Y\_5} = − 0.068 L − 0.32 l + 21.89 V$ | | 137.30 | 0.96 | 0.96 | 1.03 | <0.05 |
| $P_{PMH\_Y\_6} = 13.17 − 0.12 L − 0.33 l + 17.43 V$ | | 18.77 | 0.77 | 0.73 | 1.32 | <0.05 |
| **$P_{SMH\_Y}= 14.25·V$, m$^3$·h$^{-1}$** | **(6)** | **19.20** | **0.40** | **0.38** | **6.38** | **<0.05** |
| $P_{SMH\_Y\_1} = 34.40 − 0.30·L$ | | 29.41 | 0.62 | 0.60 | 3.72 | <0.05 |
| $P_{SMH\_Y\_2} = 14.18 V$ | | 12.91 | 0.42 | 0.39 | 9.62 | <0.05 |
| $P_{SMH\_Y\_3} = 46.15 − 0.22 L − 055 l − 0.69 V$ | | 5.15 | 0.38 | 0.30 | 5.82 | <0.05 |
| $P_{SMH\_Y\_4} = 38.95 − 0.75 l$ | | 5.61 | 0.40 | 0.33 | 5.05 | <0.05 |
| $P_{SMH\_Y\_5} = 20.72 V$ | | 11.63 | 0.39 | 0.36 | 5.10 | <0.05 |
| $P_{SMH\_Y\_6} = −0.15 L + 25.54 V$ | | 3.23 | 0.36 | 0.25 | 3.91 | <0.05 |
| **$P_{PMH\_P}= 16.58 + 21.26·V$, m$^3$·h$^{-1}$** | **(7)** | **581.26** | **0.83** | **0.83** | **3.19** | **<0.05** |
| $P_{PMH\_P\_1} = 15.98 + 23.33·V$ | | 111.62 | 0.86 | 0.85 | 3.32 | <0.05 |
| $P_{PMH\_P\_2} = 18.43 + 21.82 V$ | | 318.86 | 0.95 | 0.94 | 2.98 | <0.05 |
| $P_{PMH\_P\_3} = 19.81 + 19.76 V$ | | 152.23 | 0.89 | 0.89 | 1.92 | <0.05 |
| $P_{PMH\_P\_4} = 23.11 + 16.06 V$ | | 29.04 | 0.62 | 0.60 | 1.72 | <0.05 |
| $P_{PMH\_P\_5} = 18.11 + 18.33 V$ | | 71.80 | 0.80 | 0.79 | 1.82 | <0.05 |
| $P_{PMH\_P\_6} = 18.60 + 16.95 V$ | | 36.00 | 0.65 | 0.64 | 2.09 | <0.05 |
| **$P_{SMH\_P}= 15.03 + 21.04·V$, m$^3$·h$^{-1}$** | **(8)** | **155.56** | **0.57** | **0.57** | **6.11** | **<0.05** |
| $P_{SMH\_P\_1} = 27.54·V$ | | 24.61 | 0.580. | 0.55 | 8.31 | <0.05 |
| $P_{SMH\_P\_2} = 20.55 + 18.96 V$ | | 37.67 | 0.68 | 0.66 | 7.53 | <0.05 |
| $P_{SMH\_P\_3} = 17.54 + 19.57 V$ | | 15.85 | 0.47 | 0.44 | 5.90 | <0.05 |
| $P_{SMH\_P\_4} = 18.13 V$ | | 4.47 | 0.20 | 0.15 | 4.95 | <0.05 |
| $P_{SMH\_P\_5} = 18.34 + 17.56 V$ | | 15.56 | 0.46 | 0.43 | | <0.05 |
| $P_{SMH\_P\_6} = 21.58 V$ | | 14.05 | 0.43 | 0.39 | 4.26 | <0.05 |
| **$P_{PMH\_PTY}= 18.10 − 0.36·i − 0.05·L − 0.19·l + 10.88·V$, m$^3$·h$^{-1}$** | **(9)** | **319.61** | **0.92** | **0.91** | **1.16** | **<0.05** |
| $P_{PMH\_PTY\_1} = 11.00 − 0.062·L − 0.23 l + 10.00·V$ | | 208.41 | 0.98 | 0.97 | 0.76 | <0.05 |
| $P_{PMH\_PTY\_2} = 9.26 − 0.058 L − 0.19 l + 11.00 V$ | | 180.80 | 0.97 | 0.97 | 1.15 | <0.05 |
| $P_{PMH\_PTY\_3} = 12.63 − 0.08 L − 0.23 l + 8.81 V$ | | 69.89 | 0.93 | 0.92 | 0.86 | <0.05 |
| $P_{PMH\_PTY\_4} = − 0.011 L − 0.17 l + 12.98 V$ | | 46.64 | 0.95 | 0.88 | 0.80 | <0.05 |
| $P_{PMH\_PTY\_5} = 3.98 − 0.028 L − 0.14 l + 11.71 V$ | | 161.52 | 0.98 | 0.97 | 0.49 | <0.05 |
| $P_{PMH\_PTY\_6} = 8.23 − 0.037 L + 0.95 V$ | | 28.54 | 0.76 | 0.73 | 0.68 | <0.05 |
| **$P_{SMH\_PTY}= 7.60·V$, m$^3$·h$^{-1}$** | **(10)** | **25.02** | **0.47** | **0.45** | **2.96** | **<0.05** |
| $P_{SMH\_PTY\_1} = 12.38 − 0.11·L + 4.88·V$ | | 208.41 | 0.98 | 0.97 | 0.76 | <0.05 |
| $P_{SMH\_PTY\_2} = 6.98 V$ | | 14.08 | 0.44 | 0.41 | 4.53 | <0.05 |
| $P_{SMH\_PTY\_3} = 24.22 − 0.37 l$ | | 3.80 | 0.31 | 0.23 | 2.93 | <0.05 |
| $P_{SMH\_PTY\_4} = − 0.29 l + 9.36 V$ | | 46.64 | 0.90 | 0.88 | 0.80 | <0.05 |
| $P_{SMH\_PTY\_5} = 11.89 V$ | | 12.74 | 0.41 | 0.38 | 2.80 | <0.05 |
| $P_{SMH\_PTY\_6} = − 0.077 L + 13.43 V$ | | 5.29 | 0.50 | 0.40 | 1.77 | <0.05 |

$_{PMH}$—productive machine hour, $_{SMH}$—scheduled machine hour, $_Y$—tower yarder unit, $_P$—processor unit, $_{PTY}$—processor tower yarder.

### 3.2.2. Processor Unit

Increasing the load volume of the yarder carriage (in this case, consisting of a single tree) led to an increase in processor's performance on both a productive machine hour and scheduled machine hour basis, as described by Equations (7) and (8), respectively (Table 5). Generally, the mean processor productivity per hour at a mean load volume of 1.23 m$^3$, excluding and including delays, was estimated at 42.71 m$^3$ PMH$^{-1}$ and 40.89 m$^3$ SMH$^{-1}$ at the given operating conditions. The productivity of the processor, like the duration of its cycle, was about twice that of the yarder. Comparison to the results of thinnings in a spruce stand obtained with Mounty 4100, provided by Borz et al. [36], showed 13.158 m$^3$ PMH$^{-1}$, i.e., three times lower production rates than of the processor unit presented in this study. Again, in Borz at al. [36] studies, thinning was provided in the stand with average DBH of 15 and 21 cm, which had a significant impact on lower productivity of processing.

### 3.2.3. Processor Tower Yarder

The delay-free productivity of PTY, showed by Equation (9), will increase as the terrain slope, yarding distance and lateral yarding distance decrease, and the load volume is maximized (Table 5). The factors that affect PTY delay-free productivity are analogous to those for delay-free yarding productivity. Interestingly, the PTY productivity with delays, given by Equation (10), is rather close to that of yarder productivity with delays, and depends only on the volume of the load—in this specific case, the volume of the tree. Therefore, the productivity of PTY was mainly determined by the productivity of the yarder unit, which determines the technological process and indicators of the combined machine.

Under the given conditions, the performance of PTY can be significantly increased if more trees (probably two trees) are attached and extracted per yarder cycle, since the productivity of the processor was approximately twice that of the yarder. In comparison, the productivity rates of studied PTY of 15.20 m$^3$ PMH$^{-1}$ and 12.29 m$^3$ SMH$^{-1}$ were higher than the productivity of two PTYs in salvage logging in windfall, which were found to be 6.57 m$^3$ PSH$_{15}{}^{-1}$ for Mounty 4000 and 7.29 m$^3$ PSH$_{15}{}^{-1}$ for Koller K501, with an average diameter of harvested assortments of 27.3 cm (under bark) [23].

### 3.3. Cost Analysis

The gross costs of Syncrofalke 3t for uphill whole tree yarding in Scots pine stand were calculated at 297.48 EUR PMH$^{-1}$ (Table 6). Thus, when the studied tower yarder was productive, the extraction costs were 16.17 EUR m$^{-3}$. The increase in the productive time of a tower yarder would lead to a decrease in extraction costs.

**Table 6.** Costs characteristics of the studied processor tower yarder.

| Costs | Costs per PMH, EUR | Costs, EUR m$^{-3}$ |
|---|---|---|
| Fixed costs | 36.40 | 1.98 |
| Variable costs | 186.75 | 10.15 |
| Labor costs | 24.96 | 1.36 |
| Net costs (excluding profit) | 248.1 | 13.48 |
| Overheads and management costs | 22.33 | 1.21 |
| Gross costs (including 10% profit) | 297.48 | 16.17 |

The costs of studied PTY were two times lower compared to the costs of 32.5 ± 5.9 EUR m$^{-3}$ for Koller K507 and 36.2 ± 7.5 EUR m$^{-3}$ for Valentini V400 (both including processing at roadside) reported by Schweier et al. [37] and carried out in Germany, where labor costs are higher (costs were analyzed with data collected in various stands of different characteristics) However, the obtained costs in the presented study were twice as high, which was expected, compared to the tower cable yarder without a processor operating in the same region [35]. Therefore, the PTY used in presented stand conditions, compared

to other machines, is considered as production and cost efficient and minimally labor intensive.

In the distribution of the net costs of the studied Syncrofalke 3t PTY under these conditions, variable costs predominated (75%); they were three times higher than the sum of fixed costs (15%) and labor costs (10%). The increase in the share of variable costs of the studied PTY was due to the significantly increased prices of diesel fuel and motor, transmission and hydraulic oil, as well as other petroleum-related products. The other two groups of costs were not affected as much by the general increase in prices.

### 4. Conclusions

The productivity of PTY, 15.20 $m^3$ $PMH^{-1}$ and 12.29 $m^3$ $SMH^{-1}$ was mainly determined by the productivity of the yarder unit, which determined the technological process and indicators of the combined machine. The analysis of the delay-free work cycle of PTY showed that the most time-consuming processes were delimbing and bucking (21%), followed by outhaul and hooking (16%), load outhaul and carriage inhaul (14% each), carriage outhaul (11%), unhooking (10%), sorting, piling and clearing (8%) and directing and gripping the tree (6%). Under the given conditions, the performance of PTY significantly increased if more trees (at least two) were attached and extracted per yarder cycle, since the productivity of the processor was approximately twice that of the yarder.

The gross costs for uphill yarding of whole deciduous trees using the studied tower yarder were calculated at 297.48 EUR $PMH^{-1}$ and 16.17 EUR $m^{-3}$. The variable costs (75%) predominated in the net costs distribution, followed by the fixed costs (15%) and the labor costs (10%).

**Author Contributions:** Conceptualization, S.F.P., S.S. and A.R.P.; Methodology, S.F.P., S.S., G.A. and T.P.; writing original draft preparation, S.F.P., S.S., P.S.M. and A.R.P.; writing—review and editing, S.F.P., S.S., P.S.M. and A.R.P. All authors have read and agreed to the published version of the manuscript.

**Funding:** This research received no external funding.

**Institutional Review Board Statement:** Not applicable.

**Informed Consent Statement:** Not applicable.

**Data Availability Statement:** Not applicable.

**Acknowledgments:** This work was supported by the inter-institutional agreement between University of Forestry (Bulgaria) and the Mediterranean University of Reggio Calabria (Italy) and from the PhD course "Agricultural, Food and Forestry Science" of the Mediterranean University of Reggio Calabria (Italy).

**Conflicts of Interest:** The authors declare no conflict of interest.

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
