# Peer review of "Modeling Productivity and Estimating Costs of Processor Tower Yarder in Shelterwood Cutting of Pine Stand"

_forests, doi:10.3390/f14020195_

Round 1

Reviewer 1 Report

The manuscript was well organized to propose the equations for cycle time and productivity using the regression analyses with variables such as L, l, V, i, and n. It also calculated operation costs which were lower than existing studies. It fits the scope of this journal and should be revised with comments below:

L120: Only yarder elements were explained. Processor elements would be explained if necessary.

L161: Only variable, "i" was italic. Variables should be italic through the whole manuscript.

Figure 5: The left side, "CI" should be "CO".

More discussion on cycle time, productivity, and costs must be necessary considering different stand, machine, and operational conditions compared with existing studies.

Author Response

Dear Reviewer,

Thank you for giving us the opportunity to submit a revised version of our manuscript: Modelling productivity and estimating costs of processor tower yarder in shelterwood cutting of pine stand. We appreciate the time and effort that you have dedicated to provide and your feedback on our manuscript.

We are grateful for reviews and feedback on our paper. We have been able to incorporate changes to reflect most of the suggestions expressed in both reviews.

Reviewer 1

The manuscript was well organized to propose the equations for cycle time and productivity using the regression analyses with variables such as L, l, V, i, and n. It also calculated operation costs which were lower than existing studies. It fits the scope of this journal and should be revised with comments below:

L120: Only yarder elements were explained. Processor elements would be explained if necessary.

Respond: Thank you for this suggestion, Processor elements have been added in the text and the processor characteristics have been described in Table 3.

L161: Only variable, "i" was italic. Variables should be italic through the whole manuscript.

R: Thank you for this suggestion. The variables have been changed to italics

Figure 5: The left side, "CI" should be "CO".

R: Thank you for this suggestion. Figure 5 has been modified

More discussion on cycle time, productivity, and costs must be necessary considering different stand, machine, and operational conditions compared with existing studies.

R: There has been added more discussion and details for comparison from of other studies, which can explain why main differences (or similarity) was observed in cycle time, productivity and costs. It is all in the chapter Results and discussion presented in red (track changes) or highlighted in green. Below are copied those addings with fragments of text on top of it to understand whole paragraphs (for addings only please have a look at the main file with track changes).

However, to include it in respond section, paragraphs after addings are:

In corridors 1, 2, 3 and 6 the significant factors that determined the duration of the delay-free work cycle were the yarding distance and the lateral yarding distance, while in corridors 4 and 5 the load, i.e. the volume of the tree, also had its influence. This was probably due to less variation in load volume in corridors 1, 2, 3 and 6.

It was similar was to the work cycle percentage distribution of the Mounty 4000 PTY [20,22], even though research was carried out in spruce stands affected by outbreak of bark beetle and fungi [20]. Similar work cycle proportions when Mounty 4000 was used was also in oak stand [22], however tree trunks were processed after delimbing with chainsaw. Cross-cutting though, was recognised as difficult due to broadleaved tree species, hard in this case [22].

Generally, the mean yarding productivity per hour at mean slope yarding distance of 67.44 m and mean lateral yarding distance of 14.78 m, excluding and including delays, estimates at 23.84 m3 PMH -1 and 18.41 m3 SMH -1 at the given operating conditions. The mean yarding productivity per hour compared to the rate of tower cable yarder in salvage operation in the same region for an extraction distance of 101 m and a lateral yarding distance of 18 m, resulting in a productivity rate of 20.1 m3 PMH−1 and 12.8 m3 SMH−1 including delays, was very close [35]. That can be seen as good result in presented research with shalterwood cutting, as in salvage logging all afected trees are harvested and extracted, and in compared case it was 1/3 of them.

Comparison to the results obtained in thinnings in spruce stand with Mounty 4100 by Borz et al. [24,36] show 5.88 m3 PMH-1 at average lateral yarding distance of 22 m and yarding distance of 190.13 m, i.e. more than 3 times lower rates. This is natural, as in thinnings smaller trees are usually harvested, in this case they were of 0.308 m3 in average [35]. Also the yarding distance was nearly 3 times longer when thinning was applied: 190.13 m comparing with 67.44 m, in presented study. In order to improve the yarder productivity, the load volume (i.e. the number of trees) of the carriage is advisable to be increased, despite the increase of lateral yarding time.

Comparison to the results obtained in thinnings in spruce stand with Mounty 4100 by Borz et al. [36] showed 13.158 m3 PMH-1, i.e. 3 times lower production rates than of that of the processor unit of studied PTY presented in this study. Again, in Borz at al. [36] studies thinning was provided in the stand with average DBH of 15 and 21 cm, which had significant impact on lower productivity of processing.

The costs of studied PTY were two time lower compared to the costs of 32.5 ±5.9 € m-3 for Koller K507 and 36.2 ±7.5 € m-3 for Valentini V400 (both including processing at roadside) reported by Schweier et al. [37] and carried out in Germany, where labour costs are higher (costs were analysed with data collected in various stands of different characteristics) However, obtained costs in presented study were twice higher, which was expected, when comparing to tower cable yarder without processor operating in the same region [35]. Therefore, the PTY used in presented stand conditions, compared to other machines, is considered as production and cost efficient and minimally labor in-tensive.

In the distribution of the net costs of the studied Syncrofalke 3t PTY under these conditions variable costs (75%) predominate; they were three times higher than the sum of fixed costs (15%) and labor costs (10%). The increase in the share of variable costs of the studied PTY was due to the significantly increased prices of diesel fuel and motor, transmission and hydraulic oil, as well as other petroleum-related products. The other two groups of costs were not affected as much by the general increase of prices.

Thank you for king comments improving the manuscript. All was read many times and other editorial and text improvements have been also added for good final results!

With best regards,

Authors

Reviewer 2 Report

Dear authors,

The main drawback of the research is lack information about mounting and dismantling tower yarder lines, about time consumption and costs of that part of the whole process. After all, the manuscript deals with time study, and I am sure that lot of time was spent for that part. 

At the end of the results/discussion you comparing your results (productivity, costs) with other papers but you did not mentioned the average yarding distance of that researches. To be honest, yarding distance of your research is quite short (corridor 1 only 80 m, or average of all corridors under 150 m). It is obvious that the productivity will be high, and the costs per m3 twice low.

Almost 300 € per PMH are very high cost, but per m3 16.17 € are not so much because of high productivity at very short distances for big PTY. Comparing your costs with the cost of wood extraction with skidder will be on the side of skidder (lower costs) and my opinion is that in researched area like yours skidder will work just fine.

My final conclusion is that  in the future you should take research of PTY in the area which is more adequate for tower yarders (longer lines, bigger slopes, bigger areas...)

Kind regards

Author Response

Dear Reviewer,

Thank you for giving us the opportunity to submit a revised version of our manuscript: Modelling productivity and estimating costs of processor tower yarder in shelterwood cutting of pine stand. We appreciate the time and effort that you have dedicated to provide and your feedback on our manuscript.

We are grateful for reviews and feedback on our paper. We have been able to incorporate changes to reflect most of the suggestions expressed in both reviews.

Reviewer 2

The main drawback of the research is lack information about mounting and dismantling tower yarder lines, about time consumption and costs of that part of the whole process. After all, the manuscript deals with time study, and I am sure that lot of time was spent for that part. 

R: Thank you for that comment. Yes, it is correct when costs are main finding of the research. In presented study the interest was emphasized on modelling of productivity and finding out how this productivity can be increased and costs lowered. It was found that two trees should be extracted at one occasion for better efficiency. Costs were only visualizing actual extraction, without any additional work before prep of PTY and after extracting.

To make it more transparent, it was added in the objectives, that time of mounting and dismantling was omitted due to focus on productivity and finding factors of productivity in the end.

At the end of the results/discussion you comparing your results (productivity, costs) with other papers but you did not mentioned the average yarding distance of that researches. To be honest, yarding distance of your research is quite short (corridor 1 only 80 m, or average of all corridors under 150 m). It is obvious that the productivity will be high, and the costs per m3 twice low.

R: It was added when possible additional information about yarding distance from other studies when comparing results in presented study. All is marked in red, as the addings are in track changes mode or highlighted in green.

Almost 300 € per PMH are very high cost, but per m3 16.17 € are not so much because of high productivity at very short distances for big PTY. Comparing your costs with the cost of wood extraction with skidder will be on the side of skidder (lower costs) and my opinion is that in researched area like yours skidder will work just fine.

R: That’s correct. However, presented study on PTY was with aim to apply in any mountain conditions with not much respect to gradient slope, where skidders can’t operate. It is now highlighted more in those fragments:

In the mountain forests of Bulgaria, about 60% are on steep slopes and hence, cable yarders are particularly suitable for timber extraction.

Cable yarding systems are increasingly being used in all terrains as an alternative to conventional fully mechanized systems with harvesters and forwarders, because of their low impact on soils [4-6] and smaller dependency on slope gradient.

My final conclusion is that  in the future you should take research of PTY in the area which is more adequate for tower yarders (longer lines, bigger slopes, bigger areas...).

R: Thank you for indications. We fully agree, after presented research it makes sens to broaden findings by those factors.

Thank you for king comments improving the manuscript. All was read many times and other editorial and text improvements have been also added for good final results!

With best regards,

Authors

Round 2

Reviewer 2 Report

Dear authors,

thank you for your answers and explanation.

Kind regards